# P-Glycoprotein (ABCB1/MDR1) and BCRP (ABCG2) Limit Brain Accumulation and Cytochrome P450-3A (CYP3A) Restricts Oral Exposure of the RET Inhibitor Selpercatinib (RETEVMO)

**DOI:** 10.3390/ph14111087

**Published:** 2021-10-27

**Authors:** Yaogeng Wang, Rolf W. Sparidans, Sander Potters, Rahime Şentürk, Maria C. Lebre, Jos H. Beijnen, Alfred H. Schinkel

**Affiliations:** 1Division of Pharmacology, The Netherlands Cancer Institute, Plesmanlaan 121, 1066 CX Amsterdam, The Netherlands; y.wang@nki.nl (Y.W.); c.lebre@nki.nl (M.C.L.); j.beijnen@nki.nl (J.H.B.); 2Department of Pharmaceutical Sciences, Division of Pharmacology, Faculty of Science, Utrecht University, Universiteitsweg 99, 3584 CG Utrecht, The Netherlands; R.W.Sparidans@uu.nl (R.W.S.); Rahime_Senturk@outlook.com (R.Ş.); 3Leiden Academic Centre for Drug Research (LACDR), Faculty of Science, Leiden University, Einsteinweg 55, 2300 RA Leiden, The Netherlands; sander_potters@live.nl; 4Department of Pharmacy & Pharmacology, The Netherlands Cancer Institute, Plesmanlaan 121, 1066 CX Amsterdam, The Netherlands

**Keywords:** selpercatinib, cytochrome P450-3A, oral exposure, rearranged during transfection (RET) receptor kinase, Slco1a/1b, p-glycoprotein/ABCB1, brain accumulation

## Abstract

Selpercatinib is a targeted, FDA-approved, oral, small-molecule inhibitor for the treatment of rearranged during transfection (RET) proto-oncogene mutation-positive cancer. Using genetically modified mouse models, we investigated the roles of the multidrug efflux transporters ABCB1 and ABCG2, the OATP1A/1B uptake transporters, and the drug-metabolizing CYP3A complex in selpercatinib pharmacokinetics. Selpercatinib was efficiently transported by hABCB1 and mAbcg2, but not hABCG2, and was not a substrate of human OATP1A2, -1B1 or -1B3 in vitro. In vivo, brain and testis penetration were increased by 3.0- and 2.7-fold in *Abcb1a/1b^-/-^* mice and by 6.2- and 6.4-fold in *Abcb1a/1b;Abcg2^-/-^* mice, respectively. Oatp1a/1b deficiency did not alter selpercatinib pharmacokinetics. The ABCB1/ABCG2 inhibitor elacridar boosted selpercatinib brain penetration in wild-type mice to the levels seen in *Abcb1a/1b;Abcg2^-/-^* mice. *Cyp3a^-/-^* mice showed a 1.4-fold higher plasma AUC_0–4h_ than wild-type mice, which was then 1.6-fold decreased upon transgenic overexpression of human CYP3A4 in liver and intestine. In summary, ABCG2, and especially ABCB1, limit brain and testis penetration of selpercatinib. Elacridar coadministration could mostly reverse these effects, without causing acute toxicity. CYP3A-mediated metabolism can limit selpercatinib oral exposure and hence its tissue concentrations. These insights may be useful in the further clinical development of selpercatinib.

## 1. Introduction

The rearranged during transfection (RET) proto-oncogene encodes a receptor tyrosine kinase for members of the glial cell line-derived neurotrophic factor (GDNF) family of extracellular signaling molecules [1]. Mutations in the RET genes can lead to a number of human diseases. The loss of RET functions can irreversibly induce a syndrome characterized by intestinal obstruction known as Hirschsprung’s disease. However, mutations causing increased activity of RET functions can result in tumor formation [2]. RET tyrosine kinase receptors can be oncogenically activated by gene fusions or point mutations. RET fusions occur in different types of cancers, including lung cancers (1–2%) and papillary thyroid cancers (10–20%) [3], whereas RET mutations affect mostly medullary thyroid cancers (MTCs) [4]. Next-generation sequencing (NGS) analysis for numerous different types of patient tumors has uncovered that RET alterations can also occur in other tumor types (albeit at low frequency), including ovarian epithelial carcinoma and salivary gland adenocarcinoma [5].

Until recently, some multikinase inhibitors (MKIs) with nonselective RET-inhibitory activity have been available for patients with RET-altered cancers. For example, there were some clinical trials with cabozantinib for RET-mutant MTCs [6,7] and RET fusion-positive lung cancers [8], but results were underwhelming with considerable side effects. Similar modest activity results were found for another drug, vandetanib, in advanced or metastatic medullary thyroid cancer [9] and advanced non-small-cell lung cancer [10]. Other MKIs with potential RET activity include sunitinib, sorafenib, alectinib, nintedanib, and ponatinib. However, it is unclear if these drugs are likely to achieve improved responses compared to cabozantinib and vandetanib [11,12]. To some extent, the low activity of these inhibitors may be due to the low affinity and/or specificity for RET inhibition and substantial ‘off-target’ side effects would limit the RET-inhibition functions [13]. Thus, the limitations of these MKIs may prevent potent RET-pathway inhibition and subsequently yield poor pharmacokinetic (PK) properties and weak anti-RET positive tumor efficacy.

Unlike these nonspecific inhibitors, selpercatinib (LOXO-292, RETEVMO, Compound CID: 134436906) is a novel, highly selective, ATP-competitive small-molecule RET inhibitor, which has nanomolar potency against diverse RET alterations. In a clinical phase 1 study, it showed a 77% overall response rate in RET fusion-positive cancers, with intracranial activity and a 45% overall response rate in RET-mutant medullary thyroid cancer [14]. In May 2020, selpercatinib (RETEVMO, Eli Lilly Company) was approved by the FDA for metastatic RET fusion-positive non-small cell lung cancer (NSCLC) in adult patients and advanced or metastatic RET-mutant medullary thyroid cancer (MTC) in adult and pediatric (≥12 years old) patients [15]. However, the information on the pharmacokinetic properties of selpercatinib is still limited.

Multidrug efflux transporters of the ATP-binding cassette (ABC) protein family, especially P-glycoprotein (P-gp; ABCB1) and breast cancer resistance protein (BCRP; ABCG2), and influx transporters such as the organic anion transporting polypeptides (OATPs) can affect drug absorption, distribution, metabolism and excretion (ADME). They have a broad substrate specificity and can thus influence the safety and efficacy profiles of many specific drugs [16,17,18]. ABCB1 and ABCG2 are highly expressed in the apical membrane of epithelia in a variety of tissues, including small intestine, liver and kidney. Additionally, they are abundant in the luminal membrane of physiological barriers, such as the blood–brain barrier (BBB), blood–testis barrier (BTB) and blood–placenta barrier (BPB) [19]. Therefore, the intestinal absorption, biliary and urinary excretion and also the accumulation in the central nervous system of many antitumor drugs, including numerous tyrosine kinase inhibitors (TKIs), are restricted by ABCB1 and/or ABCG2. This interaction often results in reduced systemic exposure after oral administration (in short oral exposure) or poor brain penetration [20,21]. As brain metastases can occur in different tumor types, especially lung cancer, the potential interaction between selpercatinib and ABCB1/ABCG2 in vivo may not only limit selpercatinib oral exposure but also its brain accumulation, and thus affect therapeutic efficacy for brain metastases in lung cancer patients.

Besides the ABC efflux transporters, OATP uptake transporters, encoded by SLCO genes, are sodium-independent transmembrane uptake transporters [22,23,24,25]. With high expression in the main detoxification organ, the liver, and possibly the small intestine, both primary locations for first-pass drug metabolism, they mediate the tissue uptake of many endogenous substrates, as well as exogenous compounds, such as hormones, toxins, and numerous drugs [16,24,25,26,27]. As a member of the OATP uptake transporters, the SLCO1A/1B transporters are of particular interest considering their high expression in the liver [24] and their key roles in hepatic uptake and hence plasma clearance of several drug substrates, including may antitumor drugs [26,28,29]. Thus, it is important to investigate whether selpercatinib is a substrate of the SLCO1A/1B transporters and whether this can influence selpercatinib oral exposure and organ distribution.

The multidrug-metabolizing Cytochrome P450 3A (CYP3A) enzyme complex is the most abundant CYP enzyme in human liver, the main detoxification organ, but also in the small intestine. It therefore plays a significant role in the oxidative metabolism of approximately half of the drugs currently in clinical use. As metabolic breakdown is one of the main elimination pathways for drugs, CYP3A activity can markedly affect the plasma exposure and thus tissue levels of certain drugs [30]. Consequently, the oral exposure, therapeutic efficacy, and the potential toxicity of drugs may be influenced by the high degree of inter- and intra-individual variation that is known to occur for the CYP3A enzyme.

The primary aim of this study was to investigate ABCB1/ABCG2 and SLCO1A/1B (OATP1A/1B) transport functions in vitro by transepithelial transport and uptake assays, respectively, and clarify the in vivo impact of ABCB1/ABCG2, SLCO1A/1B and CYP3A enzymes on selpercatinib pharmacokinetic behavior, including oral exposure and organ distribution, by using appropriate genetically modified mouse models. We also further studied the effect of coadministration of the ABCB1 and ABCG2 inhibitor elacridar on selpercatinib plasma exposure and tissue distribution.

## 2. Results

### 2.1. In Vitro Transport of Selpercatinib

We tested in vitro transepithelial transport of selpercatinib using polarized monolayers of Madin-Darby Canine Kidney (MDCK-II) parental cells and its subclones overexpressing human (h) ABCB1, hABCG2, or mouse (m) Abcg2. Selpercatinib (5 µM) was not transported in the apical direction in the parental MDCK-II cell line with or without ABCB1 inhibitor zosuquidar (r = 1.0, Figure 1A and r = 0.9, Figure 1B), suggesting that selpercatinib transport could not be mediated by the low amount of endogenous canine ABCB1 present in the MDCK-II cells [31]. In MDCK-II cells transduced with hABCB1, there was clear apically directed transport of selpercatinib (r = 6.8, Figure 1C), which was completely inhibited by zosuquidar (r = 1.0, Figure 1D).

To suppress any potential confounding influence of endogenous canine ABCB1 activity, the following experiments on ABCG2-mediated transport were conducted in the presence of the inhibitor zosuquidar. In addition, the ABCG2 inhibitor Ko143 was used to inhibit the transport activity of hABCG2 and mAbcg2. In MDCK-II cells transduced with hABCG2, there was no active apically directed transport of selpercatinib (r = 1.0, Figure 1E), and this was not changed upon Ko143 addition (r = 0.9, Figure 1F). We observed strong apically directed transport of selpercatinib in cells overexpressing mouse Abcg2 (r = 8.8) and this was abrogated by addition of Ko143 (r = 1.0, Figure 1G,H).

Selpercatinib thus appears to be efficiently transported by hABCB1 and mAbcg2, but not by hABCG2 or canine ABCB1.

### 2.2. Impact of ABCB1, ABCG2 and SLCO1A/1B on Selpercatinib Plasma Pharmacokinetics and Tissue Disposition

In order to study whether the ABCB1A/1B, ABCG2 and OATP1A/1B transporters affect selpercatinib systemic exposure after oral administration (oral exposure) and subsequent tissue disposition in vivo, we performed a 4 h pharmacokinetic pilot study in male wild-type *Abcb1a/1b; Abcg2^-/-^* and *Slco1a/1b^-/-^* mice using oral administration of 10 mg/kg selpercatinib. This dose in mice results in systemic selpercatinib exposure of the same order of magnitude as seen in patients. As shown in Figure 2A,B and Table 1, after rapid initial absorption, it took around one to two hours to reach the maximum plasma concentration of selpercatinib in all tested strains, with a slow transition to elimination up to 4 h. Mice with a combined knockout of Abcb1a/1b (Mdr1) and Abcg2 (Bcrp) had a similar plasma C_max_ (8582 ± 2160 ng/mL) as wild-type mice, but mOatp1a/1b deficiency led to significantly increased selpercatinib concentrations in plasma with a 1.5-fold higher C_max_ compared to wild-type mice (11,625 ± 1614 vs. 7862 ± 1814 ng/mL, *p* < 0.05). However, plasma exposures of selpercatinib over 4 h (AUC_0–4h_) in both *Abcb1a/1b; Abcg2^-/-^* (30,188 ± 7632 ng/mL*h) and *Slco1a/1b^-/-^* (36,197 ± 5255 ng/mL*h) mice were not significantly different from those in wild-type mice (26,649 ± 6360 ng/mL*h).

Brain, liver, kidney, small intestine (SI), small intestine contents (SIC), testis, lung and spleen concentrations of selpercatinib 4 h after oral administration were analyzed. Notably, the selpercatinib brain-to-plasma ratio (0.030) in wild-type mice was quite low, suggesting poor brain penetration of selpercatinib at 4 h (Table 1). The brain concentrations and brain-to-plasma ratios in *Abcb1a/1b; Abcg2^-/-^* mice were increased by 18.6-fold and 15.3-fold, respectively, compared to those in wild-type mice (Figure 2C,D and Table 1). The *Slco1a/1b^-/-^* mice also showed enhanced brain concentrations and brain-to-plasma ratios by factors of 1.5-fold and 1.3-fold, respectively. However, these increases were quite limited compared to those in the *Abcb1a/1b; Abcg2^-/-^* mice. We further observed similar results in testis, with low testis-to-plasma ratio (0.12) in wild-type mice, a significant increase up to 0.76 (6.3-fold) in *Abcb1a/1b; Abcg2^-/-^* mice and a limited increase to 0.15 (1.3-fold) in *Slco1a/1b^-/-^* mice (Figure 2E,F and Table 1).

Whereas the other tissue-to-plasma ratios, including liver, kidney, lung and spleen, were not meaningfully altered between the three strains (liver shown in Figure 3A,B and Table 1, other data shown in Appendix A), we observed markedly lower small intestine contents-to-plasma ratios in *Abcb1a/1b/Abcg2^-/-^* mice compared to wild-type mice (0.30-fold, Figure 3E,F and Table 1). A lower small intestine contents percentage of total dose was also observed in *Abcb1a/1b/Abcg2^-/-^* mice (Figure 3G,H). These results may therefore point to more rapid absorption of intestinal selpercatinib in the absence of Abcb1a/1b and Abcg2, or to reduced hepatobiliary excretion of the absorbed selpercatinib, or to a combination of both processes.

It is worth noting that in wild-type mice, most tissue-to-plasma ratios for liver, kidney, and small intestine (all >1) were far higher than observed for the brain (0.030) and even testis (0.12), suggesting a strong impact of the blood–brain barrier (BBB) and blood–testis barrier (BTB) on tissue accumulation of selpercatinib. Despite the dramatically increased selpercatinib brain levels in *Abcb1a/1b; Abcg2^-/-^* mice, we did not observe any abnormal external behavior in these mice. This contrasts with the TKI drug brigatinib, for which we observed severe and even lethal acute toxicity in *Abcb1a/1b; Abcg2^-/-^* mice [21]. Additionally, *Slco1a/1b^-/-^* mice did not show any abnormal external behavior due to selpercatinib.

### 2.3. In Vitro Uptake of Selpercatinib

In the pilot study, we observed a relatively higher plasma concentration of selpercatinib in *Slco1a/1b^-/-^* mice. There is high expression of OATP1A/1B transporters in the liver, and they have clear effects on tissue distribution and elimination of a variety of substrates [20]. We therefore evaluated whether selpercatinib can be transported by human OATP1A2, OATP1B1 or OATP1B3 in vitro using HEK293 cells transduced with cDNAs for these transporter proteins. However, we did not observe any significant increase in the uptake of selpercatinib in any of these transgenic cell lines compared to their vector control cells (Figure 4A). Rosuvastatin, as a positive control substrate, was efficiently taken up by all the OATP-overexpressing cell lines, demonstrating that all the OATP proteins transduced in the HEK293 cells are functional (Figure 4B). Taken together, these results indicate that selpercatinib is not a substantial transport substrate of human OATP1A2, -1B1 or -1B3 as measured in HEK293 cells in vitro.

### 2.4. ABCB1 and ABCG2 Limit Selpercatinib Brain and Testis Exposure

We next performed a more extensive main experiment and studied the separate and combined functions of Abcb1a/1b and Abcg2 in selpercatinib pharmacokinetic behavior, including oral exposure and tissue distribution. In order to assess tissue distribution at a comparatively high plasma exposure, 10 mg/kg selpercatinib was administered orally to wild-type, *Abcb1a/1b^-/-^*, *Abcg2^-/-^*, and *Abcb1a/1b/Abcg2^-/-^* mice, and the experiment was terminated at 4 h. Single deficiency of either mAbcb1 or mAbcg2 resulted in higher selpercatinib plasma exposure, with the plasma AUC_0–4h_ increased in both *Abcb1a/1b^-/-^* (37,024 ± 9634 ng/mL*h, 1.6-fold, *p* < 0.01) and *Abcg2^-/-^* mice (39,056 ± 6710 ng/mL*h, 1.7-fold, *p* < 0.01) (Figure 5 and Table 2). In *Abcb1a/1b/Abcg2^-/-^* mice, the selpercatinib plasma AUC_0–4h_ was also increased up to 1.6-fold compared with wild-type mice (38,986 ± 4711 ng/mL*h vs. 23,670 ± 2469 ng/mL*h, *p* < 0.01). This result contrasts somewhat with the pilot study, where *Abcb1a/1b/Abcg2^-/-^* mice did not show a significantly higher plasma exposure than wild-type mice. This difference was mainly due to a higher plasma concentration in *Abcb1a/1b/Abcg2^-/-^* mice in the main study, where the C_max_ was increased from 8582 ng/mL in the pilot study up to 11,015 ng/mL. However, the difference between these two sets of data was small and did not alter tissue penetration effects of Abcb1 and Abcg2, especially in brain and testis.

In spite of this modest discrepancy, the single Abcb1a/1b deficiency, but not Abcg2 deficiency, profoundly increased the brain concentration by 4.8-fold compared w wild-type mice. However, this increase was even larger in mice with a combination deficiency of both Abcb1a/1b and Abcg2 (10.2-fold) compared to wild-type mice. The brain-to-plasma ratio of selpercatinib was again very low (0.077) in wild-type mice, but was increased to 0.23 (3.0-fold) by single mAbcb1 deficiency and further up to 0.48 (6.2-fold) by combined mAbcb1 and mAbcg2 deficiency (Figure 5C,D; Table 2). Notably, due to the higher brain concentration of selpercatinib in wild-type mice in the main study, this difference (6.2-fold) was lower than that observed in the pilot study (15.3-fold). Nonetheless, these results reveal that Abcb1a/1b can restrict selpercatinib brain accumulation and the further increased drug exposure in *Abcb1a/1b/Abcg2^-/-^* brains demonstrates that, in the absence of Abcb1a/1b activity, Abcg2 still limits selpercatinib brain penetration. Qualitatively similar results were obtained for selpercatinib testis penetration. Although the wild-type testis-to-plasma ratio was substantially higher (0.15) than for brain, the relative increased ratios in *Abcb1a/1b^-/-^* (2.7-fold) and *Abcb1a/1b/Abcg2^-/-^* (6.4-fold) mice were similar to those for brain (Figure 5E,F; Table 2). The data indicate that Abcb1a/1b, and to a lesser extent Abcg2, can strongly reduce the brain accumulation of selpercatinib, while testis accumulation was similarly affected.

In the main experiment, we observed significantly lower liver-to-plasma ratios in *Abcg2^-/-^* and *Abcb1a/1b/Abcg2^-/-^* mice (Figure 6A,B), which had not been obvious in *Abcb1a/1b/Abcg2^-/-^* mice in the pilot study. In theory it could be that the ABC transporters normally concentrate selpercatinib in the intrahepatic bile, and that loss of this process results in relatively reduced overall liver levels of the drug in the knockout strains. Indeed, the concentrations, the tissue-to-plasma ratios and the percentage of dose of selpercatinib in the small intestinal contents were reduced in *Abcb1a/1b^−/−^* and especially *Abcb1a/1b; Abcg2^−/−^* mice compared to wild-type mice (Figure 6 and Table 2). This finding was consistent with the pilot results and could again suggest a more rapid and extensive absorption of selpercatinib across the intestinal wall in the absence of intestinal Abcb1a/1b activity (essentially because of loss of an intestinal excretion process), or reduced hepatobiliary recirculation of absorbed selpercatinib through biliary excretion mediated by Abcb1a/1b in the bile canaliculi of the liver, or a combination of both processes. No meaningful differences were found in the other tissues analyzed (Appendix A).

### 2.5. Effect of the Dual ABCB1 and ABCG2 Inhibitor Elacridar on Selpercatinib Brain Accumulation

In view of the poor selpercatinib penetration into wild-type brain and the potential therapeutic benefit of enhancing selpercatinib brain accumulation, we investigated to what extent the dual ABCB1 and ABCG2 inhibitor elacridar could increase the brain accumulation of selpercatinib. We also assessed whether elacridar influences selpercatinib disposition and distribution in other tissues. Oral elacridar has a T_max_ of about 4 h in mice. To ensure complete inhibition of the BBB ABC transporters, elacridar (50 mg/kg) or vehicle was administered orally 2 h prior to oral selpercatinib administration (10 mg/kg) to wild-type and *Abcb1a/1b; Abcg2^−/−^* mice. Plasma and brain selpercatinib levels were assessed 2 h after selpercatinib administration. The selpercatinib plasma concentration was still high at this time point, making the impact of the BBB transporters especially relevant. In the vehicle-treated strains, the selpercatinib plasma AUC_0–2h_ was not significantly different between the strains, and pretreatment with elacridar did not result in meaningful alterations (Figure 7A,B and Table 3). In the absence of elacridar, the brain concentration and brain-to-plasma ratio of selpercatinib were 22.5-fold and 17-fold higher in *Abcb1a/1b; Abcg2^−/−^* mice than in wild-type mice, respectively (*p* < 0.001). Elacridar pretreatment markedly increased these values in wild-type mice by 13.2- and 11.5-fold, respectively (*p* < 0.001), resulting in levels close to those observed in *Abcb1a/1b; Abcg2^−/−^* mice with or without elacridar pretreatment (Figure 7C,D and Table 3). Since these parameters were not significantly altered by elacridar in *Abcb1a/1b; Abcg2^−/−^* mice, the pharmacokinetic effect of elacridar appears to be specifically mediated by the inhibition of mAbcb1a/1b and mAbcg2 in the BBB. Qualitatively similar but more modest differences were seen in testis, suggesting elacridar inhibition functions also applied in the BTB (Figure 7E,F). Unlike for the brain and testis, the liver distribution of selpercatinib was not noticeably affected by elacridar treatment in either mouse strain (Table 3). Additionally, other tissues tested did not show marked differences concerning tissue-to-plasma ratios (Appendix A).

### 2.6. Impact of CYP3A on Selpercatinib Plasma Exposure and Tissue Distribution

Many drugs and prodrugs are metabolized by CYP3A, which can therefore restrict their oral exposure. To assess the impact of CYP3A on selpercatinib in vivo, we next performed an 8 h pharmacokinetic study in female wild-type, *Cyp3a^-/-^* and Cyp3aXAV mice (with human CYP3A4 specific transgene expression in liver and intestine in a mouse Cyp3a-deficient background). Selpercatinib (10 mg/kg) was administered orally after 2–3 h of fasting, blood samples were taken at several time points and organs were collected at the last time point (8 h). The oral selpercatinib plasma AUC_0–8h_ in *Cyp3a^-/-^* mice was significantly higher (72,243 ± 5642 ng/mL*h, 1.4-fold, *p* < 0.01) than that in wild-type mice (52,251 ± 6922 ng/mL*h), while Cyp3aXAV mice showed a plasma AUC_0–8h_ (45,755 ± 3460 ng/mL*h) which was reduced again to roughly the levels seen in wild-type mice (Figure 8 and Table 4). However, regarding the tissue distribution at 8 h, the observed differences in absolute tissue concentrations for brain, liver, kidney, small intestine, testis, lung and spleen among the strains reflected the plasma AUC_0–8h_ differences, without substantial alterations in corresponding tissue-to-plasma ratios (Appendix A and Table 4). Collectively, these results indicate that selpercatinib is substantially metabolized by mouse CYP3A and human CYP3A4, which markedly affects the oral exposure, and consequently, the tissue levels of selpercatinib.

## 3. Discussion

In the current study, we found that the RET inhibitor selpercatinib is efficiently transported by human ABCB1 and mouse Abcg2 in vitro. The brain-to-plasma ratio of selpercatinib was found to be low (0.030–0.077) in wild-type mice, indicating relatively poor brain penetration of selpercatinib. This could be increased by a factor as high as 17-fold (ranging from 6.2- to 17-fold) in *Abcb1a/1b*; *Abcg2^-/-^* mice. We obtained qualitatively similar results for the impact of the ABC transporters on limiting selpercatinib testis penetration, with increases as high as 6.7-fold upon deficiency of both Abcb1 and Abcg2. Thus, our results demonstrate that ABCB1 and ABCG2 in the blood–brain barrier (BBB) could profoundly limit the brain penetration of selpercatinib, although ABCG2 showed a more modest effect. Similar functions of ABCB1 and ABCG2 also showed up in the blood–testis barrier (BTB), albeit somewhat less pronounced. Oral coadministration of the ABCB1/ABCG2 inhibitor elacridar could further mostly reverse these functions. Despite the increased plasma and tissue exposure, we did not observe any abnormal external behavior indicative of acute toxicity caused by selpercatinib in the *Abcb1a/1b*; *Abcg2^-/-^* mice (nor in the *Slco1a/1b^-/-^* and *Cyp3a^-/-^* mice). Slco1a/1b deficiencies did not significantly alter selpercatinib pharmacokinetics. Of note, at the dose used in our study (10 mg/kg), the relative pharmacokinetic parameters, including average T_max_ (~2 h), C_max_ (~8000 ng/mL) and AUC_0–8h_ (52,251 ng/mL*h) of selpercatinib in wild-type mice, were of the same order of magnitude as those observed in patients (T_max_ is 2 h with average C_max_ 2980 ng/mL and AUC_0–24h_ 51,600 ng/mL*h).

In the small intestine contents, we further observed a clearly decreased concentration and SIC-to-plasma ratio of selpercatinib in the absence of Abcb1a/1b, and these values were even lower when both Abcb1a/1b and Abcg2 were deficient (*p* < 0.01). As explained earlier, this suggests that both Abcb1a/1b and Abcg2, but mainly Abcb1a/1b, can either reduce net intestinal uptake by mediating direct efflux of selpercatinib across the intestinal wall back into the intestinal lumen, or the hepatobiliary excretion of selpercatinib, or a combination of both processes. No notable changes in tissue distribution were observed in other tissues due to the ABC transporter deficiencies, including liver, kidney, lung and spleen.

Oncogenic RET fusions occur infrequently in diverse types of cancer, including NSCLC (1–2%), and more frequently in papillary thyroid cancers (10–20%). Frequencies in other rare solid tumors are even lower. Although infrequent, RET fusions appear to be associated with a high risk of brain metastases, which was demonstrated by the finding that the cumulative incidence of CNS lesions in RET-positive NSCLC patients is higher than that in ROS1-positive patients [32]. Thus, it is worthwhile to investigate whether selpercatinib can achieve high intrinsic BBB permeability, and the potential effects due to interaction of selpercatinib with ABCB1 and ABCG2 in the BBB. While this project was ongoing, the FDA approved selpercatinib (Food and Drug Administration, 2020) [15]. According to its guidelines, selpercatinib is a substrate of ABCB1 and ABCG2. However, in our in vitro results, selpercatinib was a good substrate of human ABCB1 and mouse Abcg2, but not of human ABCG2. Despite this, the guidelines appear in accordance with our in vivo data, especially in the BBB and BTB, where ABCB1 displayed a main protective function, while ABCG2 had a smaller effect.

The observed strong interactions of selpercatinib with ABCB1 and ABCG2 could well result in poor brain penetration in humans too, potentially limiting therapeutic efficacy. So far, there is little direct documentation about human selpercatinib brain penetration or accumulation. Drilon et al. (2020) reported that selpercatinib was designed to penetrate the central nervous system (CNS) and had been shown in preclinical models to have antitumor activity in the brain. In the Phase 1–2 clinical trial, 38 of 105 patients had investigator-assessed CNS metastasis at baseline and 11 patients were deemed to have measurable lesions. Among these 11 patients, the percentage with an objective intracranial response was 91% (10 of 11 patients; 95% CI, 59 to 100) according to independent review, including 3 complete responses (27%), 7 partial responses (64%), and 1 stable disease [33]. However, our results show that selpercatinib indeed has a poor brain penetration in wild-type mice, mainly due to the activity of ABCB1 in the BBB. This ABCB1 P-glycoprotein function may be of relevance for further increasing therapeutic efficacy against brain metastases in RET-mutated NSCLC, in case ABCB1 in the human brain has a similar impact as in the mouse brain. If so, looking ahead for a broader clinical use of selpercatinib, we could also use this insight to improve (boost) brain concentration of selpercatinib using pharmacological inhibitors of P-glycoprotein, such as elacridar. From our results, oral co-administration of elacridar did not alter the overall plasma exposure of selpercatinib. Importantly, however, brain distribution of selpercatinib was profoundly improved in wild-type mice by elacridar (from 0.041 to 0.46, 11.5-fold) without any abnormal external behavior, albeit not to as high a level as seen in vehicle-treated *Abcb1a/1b; Abcg2^-/-^* mice (0.68, 17.0-fold). We thus demonstrated that extensive inhibition of Abcb1 in the BBB could be achieved using a clinically realistic coadministration schedule.

However, more drug accumulation in the brain may also induce CNS toxicity, as we observed in a previous study with the ALK/EGFR inhibitor brigatinib. We found sometimes severe and lethal toxicity of oral brigatinib in mice with genetic knockout or pharmacological inhibition of mAbcb1a/1b and mAbcg2 [21]. Related to this, recently selpercatinib was also being investigated in combination with previously registered anticancer drugs such as crizotinib in patients with RET-positive NSCLC to overcome MET mutated resistance [34]. According to the FDA documentation and our previous study [35,36], crizotinib is a substrate and also an inhibitor of ABCB1. Given the marked drug–drug interaction between selpercatinib and elacridar, any attempt to apply efficacious ABCB1/ABCG2 inhibitors in patients together with selpercatinib should be carefully monitored, even though no noticeable signs of acute selpercatinib CNS toxicity were observed in our study.

We further found that selpercatinib oral exposure in mice was modestly restricted by mouse Cyp3a (1.4-fold) and similarly by human CYP3A4 when compared to wild-type mice. This demonstrates that the metabolic clearance of selpercatinib is substantially influenced by human CYP3A4. Despite the clear differences in tissue concentrations among mouse strains, we did not observe any meaningful changes in corresponding tissue-to-plasma ratios. This suggests that mouse Cyp3a and human CYP3A4 have little effect on relative selpercatinib tissue distribution and that the absolute tissue concentration differences only reflected the plasma concentration differences among the mouse strains. Consistent with the FDA declaration, our results indicate a clear in vivo interaction of selpercatinib and CYP3A. Thus, the body exposure and metabolic clearance of selpercatinib would likely be noticeably affected by variable CYP3A activity in patients, due to either drug–drug interactions or genetic polymorphisms, potentially compromising its therapeutic effect and safety. This further emphasizes the importance of critically monitoring clinical dosing of selpercatinib due to individual CYP3A activity variation and/or when administering selpercatinib together with CYP3A inducers and/or inhibitors.

Our study provides insights into in vivo functions of detoxifying systems with respect to selpercatinib pharmacokinetics. Still, it is useful to keep in mind that while mouse studies can be used to obtain qualitative insights into the functions of various detoxifying proteins, their exact impact in patients always needs to be investigated in their own right in human studies.

## 4. Materials and Methods

### 4.1. Cell Lines and Transport Assays

Polarized Madin-Darby Canine Kidney (MDCK-II) cell lines and subclones stably transduced with either human (h) ABCB1, hABCG2, or mouse (m) Abcg2 cDNA were generated in The Netherlands cancer institute between 1995 and 2005. The characteristic growth and drug transport properties of the cell lines, including inhibitor sensitivity, were regularly checked. They are continually used in our recent studies, and the proper identity and functionality of these polarized epithelial cells has been confirmed [20,37]. The transepithelial transport experiments were performed as described previously [20]. Briefly, transepithelial transport assays were performed on microporous polycarbonate membrane filters (3.0 µm pore size, 12 mm diameter, Transwell 3402, Corning, NY, USA). The parental cells and subclones were seeded at a density of 2.5 × 10^5^ cells per well and cultured for 3 days to allow formation of an intact monolayer. Membrane tightness was assessed by measurement of transepithelial electrical resistance (TEER) before and after the transport phase.

In the inhibition experiments, 5 μM zosuquidar (ABCB1 inhibitor) and/or 5 μM Ko143 (ABCG2/Abcg2 inhibitor) were used during the transport experiments. Cells were preincubated with one or a combination of the inhibitors for 1 h in both apical and basolateral compartments. The transport phase was started (t = 0) by replacing the medium in either the apical or the basolateral compartment with fresh DMEM including 10% (*v*/*v*) fetal bovine serum (FBS) and selpercatinib at 5 μM, as well as the appropriate inhibitor (s). Plates then were kept at 37 °C in 5% (*v*/*v*) CO_2_ during the experiment, and 50 μL aliquots were taken from the acceptor compartment at 1, 2, 4, and 8 h (h), and stored at −30 °C until LC–MS/MS measurement of the selpercatinib concentrations. Experiments were performed in triplicate and the mean transport is shown in the figure. Active transport was expressed using the transport ratio (r), which is defined as the amount of apically directed drug transport divided by basolaterally directed drug translocation after 8 h.

### 4.2. Cellular Uptake Assays

HEK293 cells, vector-transfected or human (h)SLCO1A2, hSLCO1B1 or hSLCO1B3 cDNA-transfected were kind gifts from Prof. Werner Siegmund and Dr. Markus Keiser (University of Greifswald, Greifswald, Germany) [38]. All the HEK293 cell lines were maintained in DMEM supplemented with 10% (*v*/*v*) FBS, 1% penicillin–streptomycin mix at 37 °C in 5% (*v*/*v*) CO_2_ and 500 μg/mL G418. Cells were first seeded in 12-well plates [coated with 50 mg/L poly (l-lysine) and 50 mg/L poly (l-ornithine)] at a density of 1.0 × 10^5^ cells/well. For the uptake study, in order to induce the expression of OATP transporters, the cell culture medium was replaced with culture medium supplemented with 5 mM sodium butyrate 24 h before performing the uptake assay.

The uptake transport study was performed as described previously [20]. Briefly, cells were first washed twice and preincubated with Krebs–Henseleit solution at 37 °C for 15 min, then uptake was initiated by adding Krebs–Henseleit buffer containing 5 μM selpercatinib or 0.2 μM rosuvastatin as a positive control. The Krebs–Henseleit solution was prepared from Krebs–Henseleit-buffer modified powder and supplemented with 25 mM NaHCO_3_ and 2.5 mM CaCl_2_ adjusted to pH 6.4 with 1 M HCL. At 2.5 min, the incubation buffer was removed, and uptake was terminated by adding 1 mL of ice-cold Krebs–Henseleit buffer, followed by two times washing with 1 mL of ice-cold Krebs–Henseleit buffer. Afterwards, cells were lysed with 150 μL of 0.2 N NaOH for 15 min at room temperature, and cell lysates were transferred into 1.5 mL Eppendorf tubes and stored at −30 °C until the next day. The cellular protein amount was determined by the Bradford method using 10 μL of the cell lysates with bovine serum albumin as a standard. LC–MS/MS measurements of the selpercatinib and rosuvastatin concentrations were performed for cell lysates. Experiments were performed in independent triplicates and the mean transport is shown in the figure. Similarly to in the transepithelial transport assay, n = 3 was considered sufficient for the selpercatinib and rosuvastatin uptake results.

### 4.3. Animals

Mice were housed and handled according to institutional guidelines complying with Dutch and EU legislation. All experimental animal protocols were evaluated and approved by the institutional animal care and use committee. Wild-type, *Abcb1a/1b^-/-^*, *Abcg2^-/-^*, *Abcb1a/1b; Abcg2^-/-^*, *Slco1a/1b^-/-^* male mice, and *Cyp3a^-/-^* and Cyp3aXAV female mice, all of a >99% FVB genetic background, were used between 9 and 16 weeks of age. These mouse strains are continually used for pharmacokinetic studies with various drugs, confirming their proper identity and functionality [20,39] and ongoing studies. Animals were kept in a temperature-controlled environment with a 12 h light and 12 h dark cycle, and they received a standard diet (Transbreed, SDS Diets, Technilab-BMI, Someren, The Netherlands) and acidified water ad libitum. All experimental animal protocols (WP9450, 9669, 9760), including power calculations, designed under the nationally approved DEC/CCD project AVD301002016595, were evaluated and approved by the institutional animal care and use committee of The Netherlands Cancer Institute.

### 4.4. Drug Solutions

For oral administration, selpercatinib was dissolved in dimethyl sulfoxide (DMSO) at a concentration of 20 mg/mL and further diluted with polysorbate 20, 100% ethanol and 5% glucose water, resulting in a final working solution of 1 mg/mL [DMSO: polysorbate 20: 100% ethanol: 5% glucose water = 5:15:15:65, (*v*/*v*/*v*/*v*)]. Elacridar hydrochloride was dissolved in DMSO (53 mg/mL) in order to obtain 50 mg elacridar base per mL DMSO. The stock solution was further diluted with a mixture of polysorbate 20, 100% ethanol and 5% glucose water to yield a concentration of 5 mg/mL elacridar [DMSO: polysorbate 20: 100% ethanol: 5% glucose water = 10:15:15:60, (*v*/*v*/*v*/*v*)]. All dosing solutions were prepared freshly on the day of the experiment.

### 4.5. Plasma and Organ Pharmacokinetics of Selpercatinib in Ice

In order to minimize variation in absorption due to of oral administration, mice were first fasted for 3 h before selpercatinib (10 mg/kg) was administered orally, using a blunt-ended needle. For the 4 h transporter experiments, tail-vein blood samples were collected at 0.125, 0.25, 0.5, 1, and 2 h time points after oral administration, respectively. For the CYP3A 8 h experiments, tail-vein blood sampling was performed at 0.25, 0.5, 1, 2, and 4 h, respectively. While for elacridar inhibition experiments, tail-vein blood samples were collected at 0.125, 0.25, 0.5, and 1 h time points after oral administration, respectively. Blood sample collection was performed using microvettes containing dipotassium EDTA. At the last time point in each experiment (2, 4 or 8 h), mice were anesthetized with 5% isoflurane and blood was collected by cardiac puncture. Blood samples were collected in Eppendorf tubes containing heparin as an anticoagulant. The mice were then sacrificed by cervical dislocation and brain, liver, kidney, lung, small intestine and testis were rapidly removed. Plasma was isolated from the blood by centrifugation at 9000× *g* for 6 min at 4 °C, and the plasma fraction was collected and stored at −30 °C until analysis. Organs were homogenized with 4% (*w*/*v*) bovine serum albumin and stored at −30 °C until analysis. Relative tissue-to-plasma ratio after oral administration was calculated by determining the selpercatinib tissue concentration relative to selpercatinib plasma concentration at the last time point.

### 4.6. LC–MS/MS Analysis

Selpercatinib concentrations in DMEM/FBS (9/1, *v*/*v*) (Invitrogen, Waltham, MA, USA) cell culture medium, plasma samples, and organ homogenates were determined using a validated liquid chromatography–tandem mass spectrometry assay [40].

### 4.7. Materials

Selpercatinib was purchased from Chemgood (Glen Allen, VA, USA). Zosuquidar and elacridar HCl were obtained from Sequoia Research Products (Pangbourne, UK). Ko143 was from Tocris Bioscience (Bristol, UK). Bovine Serum Albumin (BSA) Fraction V was obtained from Roche Diagnostics GmbH (Mannheim, Germany). Glucose water 5% *w/v* was from B. Braun Medical Supplies, Inc. (Melsungen, Germany). Isoflurane was purchased from Pharmachemie (Haarlem, The Netherlands) and heparin (5000 IU mL^−1^) was from Leo Pharma (Breda, The Netherlands). Other chemicals used in the selpercatinib detection assay were described before [40]. All other chemicals and reagents were obtained from Sigma-Aldrich (Steinheim, Germany).

### 4.8. Data and Statistical Analysis

Pharmacokinetic parameters were calculated by noncompartmental methods using the PK solver software [41]. The area under the plasma concentration–time curve (AUC) was calculated using the trapezoidal rule, without extrapolating to infinity. The peak plasma concentration (C_max_) and the time of maximum plasma concentration (T_max_) were estimated from the original (individual mouse) data. One-way analysis of variance (ANOVA) was used when multiple groups were compared and the Tukey *post hoc* correction was used to accommodate multiple testing. The two-sided unpaired Student’s t-test was used when treatments or differences between two specific groups were compared using the software GraphPad Prism 8 (GraphPad Software Inc., La Jolla, CA, USA). All the data were log-transformed before statistical tests were applied. Differences were considered statistically significant when *p* < 0.05. All data are presented as geometric mean ± SD.

## 5. Conclusions

In summary, ABCG2 and especially ABCB1 can limit the oral exposure and brain and testis penetration of selpercatinib, as well as its intestinal disposition. To the best of our knowledge, this is the first study documenting that elacridar can improve selpercatinib brain accumulation. Additionally, CYP3A-mediated metabolism can markedly reduce selpercatinib oral exposure and thus its tissue concentrations. The obtained insights and principles may potentially be used to further enhance the therapeutic application and efficacy of selpercatinib, especially for brain metastases in RET fusion/mutation-positive NSCLC patients.

## Figures and Tables

**Figure 1 pharmaceuticals-14-01087-f001:**
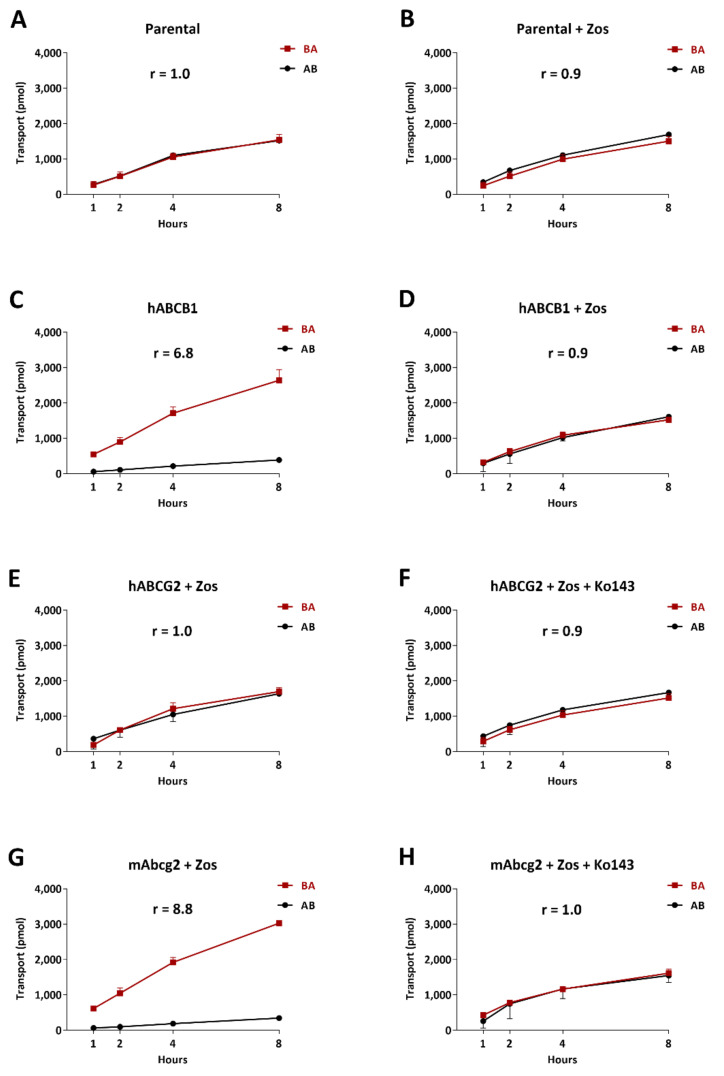
Transepithelial transport of selpercatinib (5 µM) assessed in MDCK-II cells either nontransduced (**A**,**B**), transduced with hABCB1 (**C**,**D**), hABCG2 (**E**,**F**) or mAbcg2 (**G**,**H**) cDNA. At *t* = 0 h, drug was applied in the donor compartment and the concentrations in the acceptor compartment at *t* = 1, 2, 4 and 8 h were measured and plotted as cumulative amount of selpercatinib transported per well (pmol) in the graphs (n = 3). (**B**,**D**–**H**): Zosuquidar (Zos, 5 μM) was applied to inhibit human and/or endogenous canine ABCB1. (**F**,**H**): the ABCG2 inhibitor Ko143 (5 μM) was applied to inhibit ABCG2/Abcg2-mediated transport. *r*, relative transport ratio. AB (●), translocation from the apical to the basolateral compartment; BA (■), translocation from the basolateral to the apical compartment. Points, mean; bars, S.D.

**Figure 2 pharmaceuticals-14-01087-f002:**
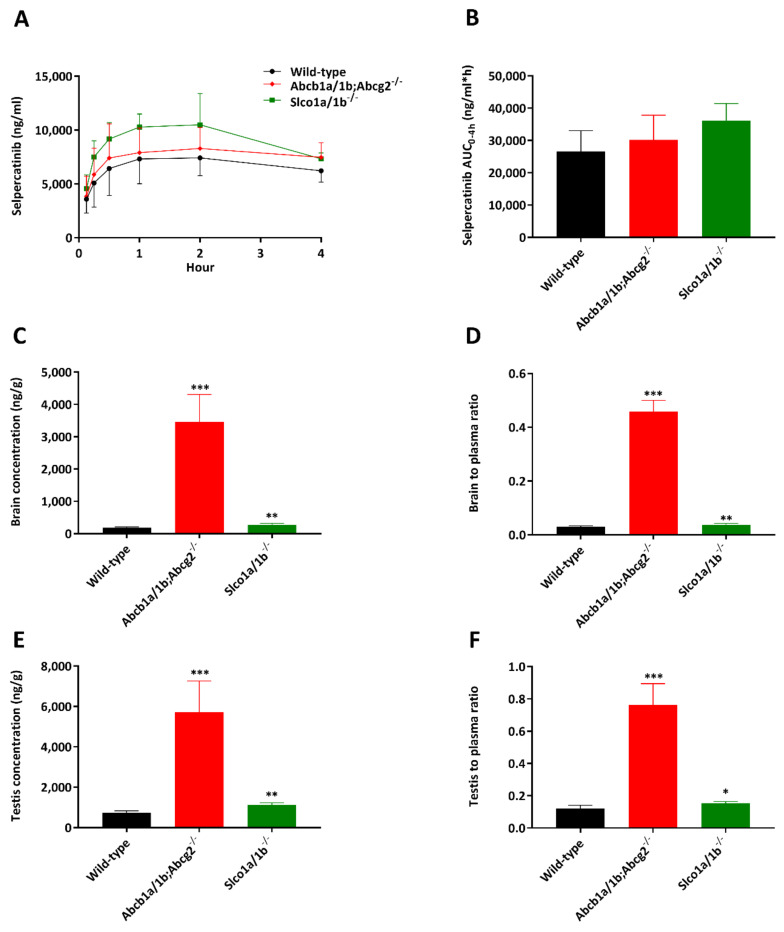
Plasma concentration–time curves (**A**), plasma AUC_0–4h_ (**B**), brain concentration (**C**), brain-to-plasma ratio (**D**), testis concentration (**E**) and testis-to-plasma ratio (**F**) of selpercatinib in male wild-type, *Abcb1a/1b; Abcg2^-/-^* and *Slco1a/1b^-/-^*mice over 4 h after oral administration of 10 mg/kg selpercatinib. Data are given as mean ± S.D. (n = 6–7). Statistical analysis was applied after log-transformation of linear data. *, *p* < 0.05; **, *p* < 0.01; ***, *p* < 0.001 compared to wild-type mice. Statistical analysis was applied after log-transformation of linear data.

**Figure 3 pharmaceuticals-14-01087-f003:**
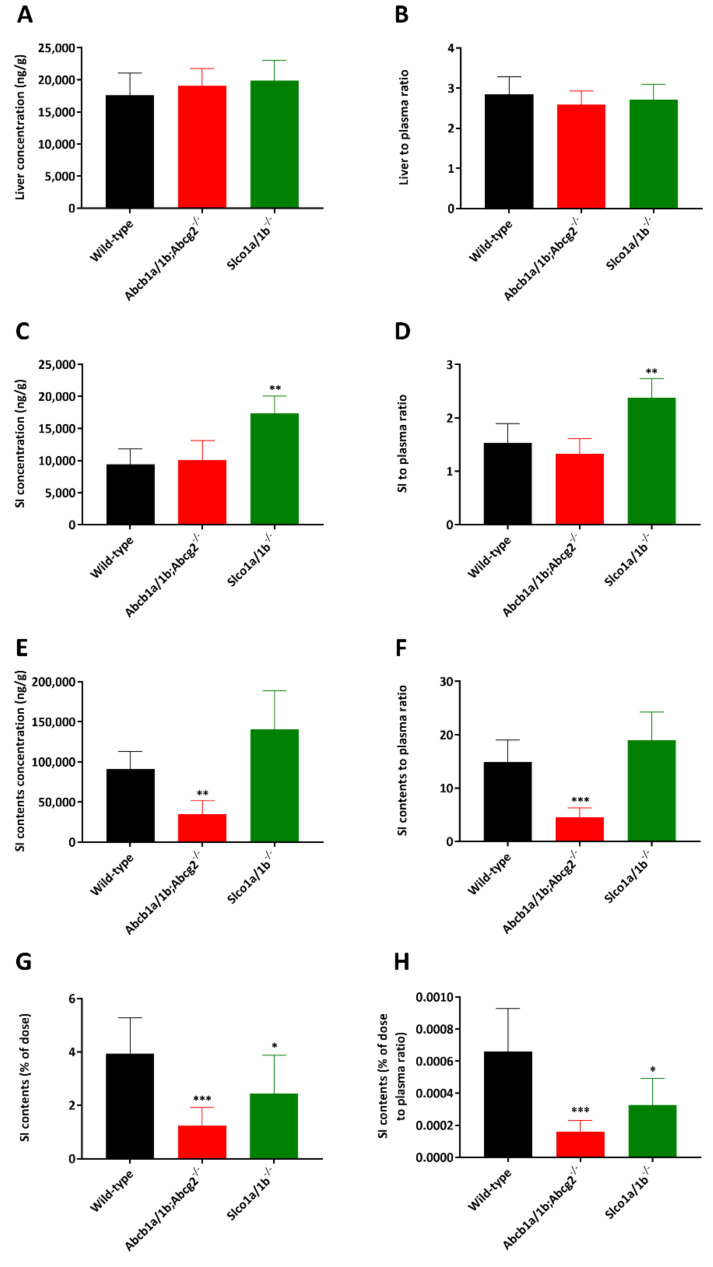
Liver, small intestine and small intestine contents concentrations (**A**,**C**,**E**), liver, small intestine- and small intestine contents-to-plasma ratios (**B**,**D**,**F**), small intestine contents as percentage of dose (**G**) and small intestine contents percentage of dose-to-plasma ratio (**H**) of selpercatinib in male wild-type, *Abcb1a/1b; Abcg2^-/-^* and *Slco1a/1b^-/-^* mice over 4 h after oral administration of 10 mg/kg selpercatinib. SI: small intestine. Data are given as mean ± S.D. (n = 6–7). *, *p* < 0.05; **, *p* < 0.01; ***, *p* < 0.001 compared to wild-type mice. Statistical analysis was applied after log-transformation of linear data.

**Figure 4 pharmaceuticals-14-01087-f004:**
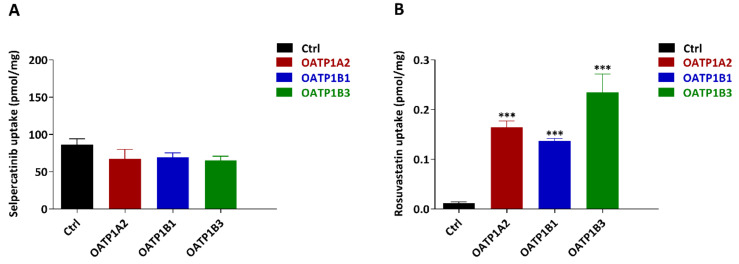
In vitro uptake of selpercatinib (**A**) and positive control rosuvastatin (**B**) by human OATP1A2, OATP1B1 and OATP1B3. Uptake of 5 µM selpercatinib and 0.2 µM rosuvastatin were measured after 2.5 min incubation using vector-transfected (control) or OATP1A2-, OATP1B1- or OATP1B3- overexpressing HEK293 cells. n = 3, data are given as mean ± S.D. *, *p* < 0.05; **, *p* < 0.01; ***, *p* < 0.001 compared to control group.

**Figure 5 pharmaceuticals-14-01087-f005:**
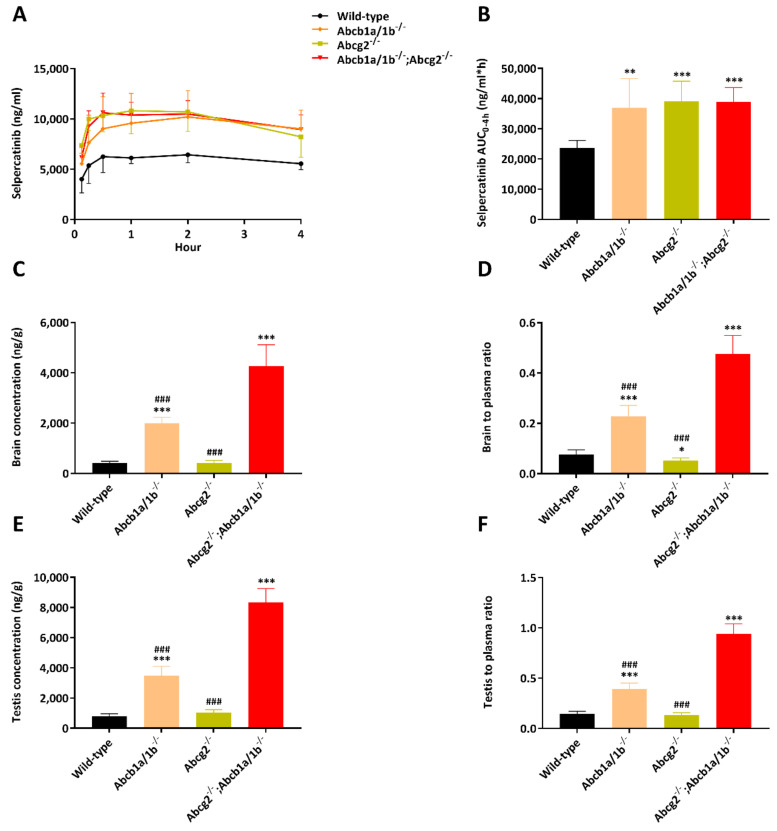
Plasma concentration–time curves (**A**), plasma AUC_0–4h_ (**B**), brain concentration (**C**), brain-to-plasma ratio (**D**), testis concentration (**E**) and testis-to-plasma ratio (**F**) of selpercatinib in male wild-type, *Abcb1a/1b^-/-^, Abcg2^-/-^* and *Abcb1a/1b; Abcg2^-/-^* mice over 4 h after oral administration of 10 mg/kg selpercatinib. Data are given as mean ± S.D. (n = 6). *, *p* < 0.05; **, *p* < 0.01; ***, *p* < 0.001 compared to wild-type mice; #, *p* < 0.05; ##, *p* < 0.01; ###, *p* < 0.001 compared to *Abcb1a/1b; Abcg2^-/-^* mice. Statistical analysis was applied after log-transformation of linear data.

**Figure 6 pharmaceuticals-14-01087-f006:**
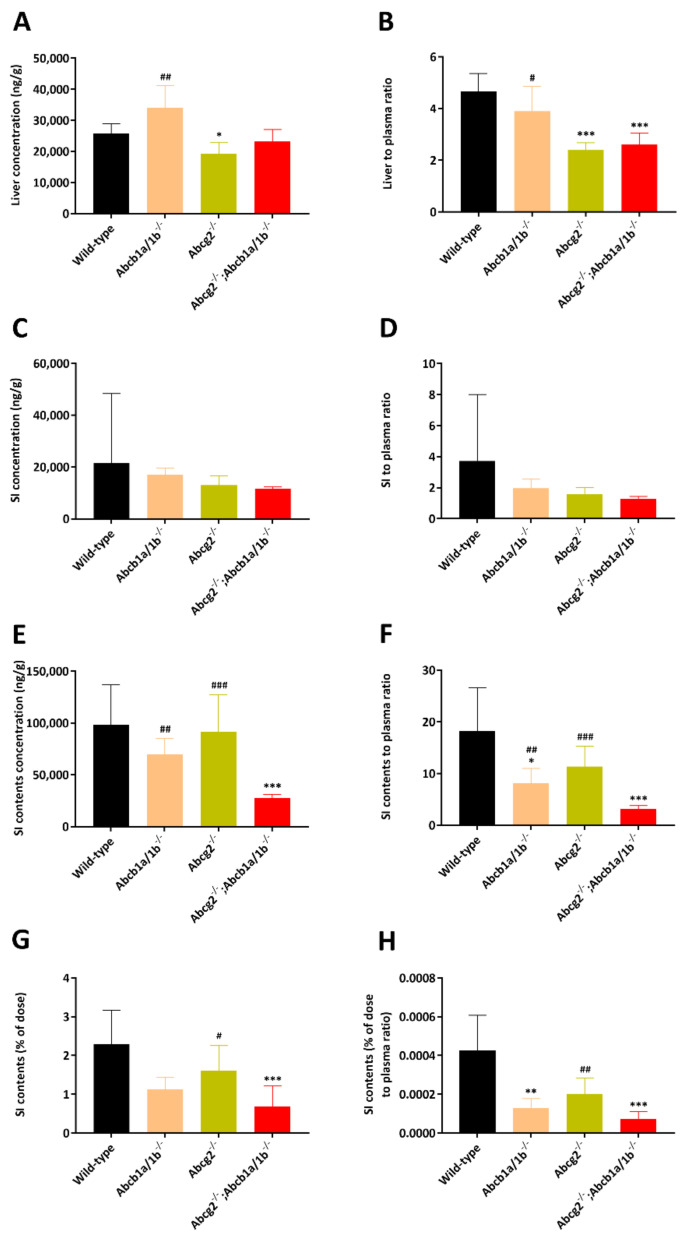
Liver, small intestine and small intestine contents concentrations (**A**,**C**,**E**), liver, small intestine- and small intestine contents-to-plasma ratios (**B**,**D**,**F**), small intestine contents as percentage of dose (**G**) and small intestine contents percentage of dose-to-plasma ratio (**H**) of selpercatinib in male wild-type, *Abcb1a/1b^-/-^, Abcg2^-/-^* and *Abcb1a/1b; Abcg2^-/-^* mice 4 h after oral administration of 10 mg/kg selpercatinib. SI: small intestine. Data are given as mean ± S.D. (n = 6). *, *p* < 0.05; **, *p* < 0.01; ***, *p* < 0.001 compared to wild-type mice; #, *p* < 0.05; ##, *p* < 0.01; ###, *p* < 0.001 compared between *Abcb1a/1b;Abcg2^-/-^* and *Slco1a/1b^-/-^* mice. Statistical analysis was applied after log-transformation of linear data.

**Figure 7 pharmaceuticals-14-01087-f007:**
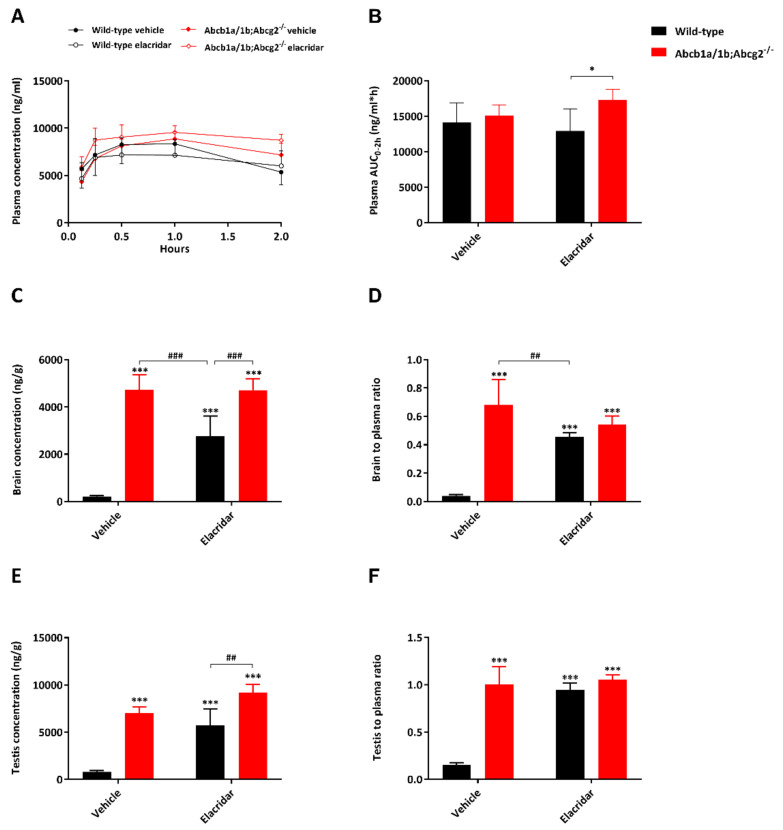
Plasma concentration–time curves (**A**), plasma AUC_0–4h_ (**B**), brain concentration (**C**), brain-to-plasma ratio (**D**), testis concentration (**E**) and testis-to-plasma ratio (**F**) of selpercatinib in male wild-type and *Abcb1a/1b;Abcg2^-/-^* mice over 2 h after oral administration of 10 mg/kg selpercatinib with or without coadministration of elacridar. Data are given as mean ± S.D. (n = 6). *, *p* < 0.05; **, *p* < 0.01; ***, *p* < 0.001 compared to vehicle-treated wild-type mice; #, *p* < 0.05; ##, *p* < 0.01; ###, *p* < 0.001 compared to elacridar-treated wild-type mice. Statistical analysis was applied after log-transformation of linear data.

**Figure 8 pharmaceuticals-14-01087-f008:**
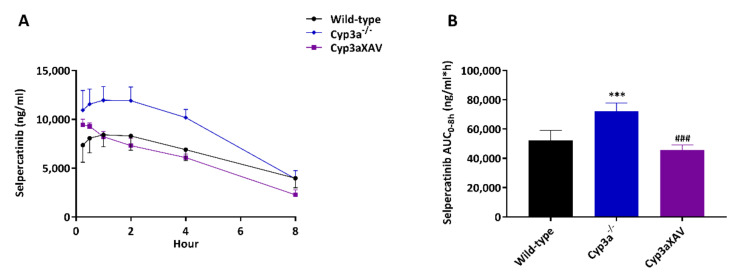
Plasma concentration–time curves (**A**) and plasma AUC_0–8h_ (**B**) of selpercatinib in female wild-type, *Cyp3a^-/-^* and Cyp3aXAV mice 8 h after oral administration of 10 mg/kg selpercatinib. Data are given as mean ± S.D. (n = 6–7). *, *p* < 0.05; **, *p* < 0.01; ***, *p* < 0.001 compared to wild-type mice; #, *p* < 0.05; ##, *p* < 0.01; ###, *p* < 0.001 compared between *Cyp3a^-/-^* and Cyp3aXAV mice. Statistical analysis was applied after log-transformation of linear data.

**Table 1 pharmaceuticals-14-01087-t001:** Plasma and organ pharmacokinetic parameters of selpercatinib in male wild-type, *Abcb1a/1b; Abcg2^-/-^* and *Slco1a/1b^-/-^* mice over 4 h after oral administration of 10 mg/kg selpercatinib.

Parameter	Genotype		
	Wild-Type	*Abcb1a/1b;Abcg2^-/-^*	*Slco1a/1b^-/-^*
AUC_0–4h_, ng/mL*h	26,649 ± 6360	30,188 ± 7632	36,197 ± 5255
Fold change AUC_0–4h_	1.0	1.1	1.4
C_max_, ng/mL	7862 ± 1814	8582 ± 2160	11,625 ± 1614 *
T_max_, h	1.8 ± 1.2	1.6 ± 1.2	1.7 ± 0.52
C_brain_, ng/g	186 ± 23	3454 ± 855 ***	278 ± 42 **
Fold increase C_brain_	1.0	18.6	1.5
Brain-to-plasma ratio	0.030 ± 0.004	0.46 ± 0.04 ***	0.038 ± 0.005 **
Fold increase ratio	1.0	15.3	1.3
C_liver_, ng/g	17,593 ± 3471	19,077 ± 2696	19,916 ± 3174
Fold increase C_liver_	1.0	1.1	1.1
Liver-to-plasma ratio	2.8 ± 0.4	2.6 ± 0.3	2.7 ± 0.4
Fold increase ratio	1.0	0.93	1.0
C_SIC_, ng/g	91,051 ± 22,029	34,929 ± 16,659 *	140,998 ± 48,076
Fold change C_SIC_	1.0	0.38	1.5
SIC-to-plasma ratio	14.9 ± 4.1	4.5 ± 1.8 ***	19.0 ± 5.3
Fold increase ratio	1.0	0.30	1.3
C_testis_, ng/g	730 ± 103	5726 ± 1535 ***	1121 ± 107 **
Fold increase C_testis_	1.0	7.8	1.5
Testis-to-plasma ratio	0.12 ± 0.02	0.76 ± 0.13 ***	0.15 ± 0.01 *
Fold increase ratio	1.0	6.3	1.3

AUC_0–4h_, area under plasma concentration–time curve; C_max_, maximum concentration in plasma; T_max_, time point (h) of maximum plasma concentration; C_brain_, brain concentration; C_liver_, liver concentration; SIC, small intestine contents; C_SIC_, small intestine contents concentration; C_testis_, testis concentration; Data are given as mean ± S.D. (n = 6–7). *, *p* < 0.05; **, *p* < 0.01; ***, *p* < 0.001 compared to wild-type mice. Statistical analysis was applied after log-transformation of linear data.

**Table 2 pharmaceuticals-14-01087-t002:** Plasma and organ pharmacokinetic parameters of selpercatinib in male wild-type, *Abcb1a/1b ^-/-^, Abcg2^-/-^* and *Abcb1a/1b; Abcg2^-/-^* mice over 4 h after oral administration of 10 mg/kg selpercatinib.

Parameter	Genotype			
	Wild-Type	*Abcb1a/1b^-/-^*	*Abcg2^-/-^*	*Abcb1a/1b*;*Abcg2^-/-^*
AUC_0–4h_, ng/mL*h	23,670 ± 2469	37,024 ± 9634 **	39,056 ± 6710 ***	38,986 ± 4711 ***
Fold change AUC_0–4h_	1.0	1.6	1.7	1.6
C_max_, ng/mL	6908 ± 761	10,338 ± 2641 **	11,447 ± 2155 ***	11,015 ± 1653 **
T_max_, h	1.6 ± 1.4	1.8 ± 1.3	1.4 ± 0.74	1.5 ± 1.3
C_brain_, ng/g	420 ± 66	1998 ± 233 ***###	418 ± 102 ###	4263 ± 853 ***
Fold change C_brain_	1.0	4.8	1.0	10.2
Brain-to-plasma ratio	0.077 ± 0.018	0.23 ± 0.04 ***###	0.052 ± 0.011 *###	0.48 ± 0.07 ***
Fold change ratio	1.0	3.0	0.68	6.2
C_Liver_, ng/g	25,737 ± 3219	34,014 ± 7169 ##	19,334 ± 3483 *	23,157 ± 3900
Fold increase C_liver_	1.0	1.3	0.75	0.90
Liver-to-plasma ratio	4.7 ± 0.7	3.9 ± 1.0 #	2.4 ± 0.3 ***	2.6 ± 0.5 ***
Fold change ratio	1.0	0.83	0.51	0.55
C_SI_, ng/g	21,649 ± 26,731	16,952 ± 2729	12,967 ± 3639	11,576 ± 734
Fold increase C_SI_	1.0	0.83	0.60	0.53
SI-to-plasma ratio	3.7 ± 4.3	2.0 ± 0.6	1.6 ± 0.4	1.3 ± 0.1
Fold change ratio	1.0	0.54	0.43	0.35
C_SIC_, ng/g	98,083 ± 38,906	69,396 ± 15,655 ##	91,686 ± 35,700 ###	27,386 ± 3533 ***
Fold increase C_SIC_	1.0	0.71	0.93	0.28
SIC-to-plasma ratio	18.2 ± 8.5	8.1 ± 2.9 *##	11.4 ± 3.9 ###	3.1 ± 0.68 ***
Fold change ratio	1.0	0.45	0.63	0.17
SIC percentage of dose, %	2.3 ± 0.9	1.1 ± 0.3	1.6 ± 0.7 #	0.69 ± 0.52 ***
Fold change ratio	1.0	0.48	0.70	0.30
C_testis_, ng/g	810.8 ± 148.4	3477 ± 634.1 ***###	1049 ± 192.2 ###	8328 ± 934.8 ***
Fold change C_testis_	1.0	4.3	1.3	10.3
Testis-to-plasma ratio	0.15 ± 0.02	0.39 ± 0.06 ***###	0.13 ± 0.03 ###	0.94 ± 0.10 ***
Fold change ratio	1.0	2.7	0.90	6.4

Data are given as mean ± S.D. (n = 6). AUC_0–4h_, area under the plasma concentration–time curve; C_max_, maximum concentration in plasma; T_max_, time point (h) of maximum plasma concentration; C_brain_, brain concentration; C_liver_, liver concentration; SI, small intestine (tissue); C_SI_, small intestine tissue concentration; SIC, small intestine contents; C_SIC_, small intestine contents concentration; C_testis_, testis concentration; *, *p* < 0.05; **, *p* < 0.01; ***, *p* < 0.001 compared to wild-type mice; #, *p* < 0.05; ##, *p* < 0.01; ###, *p* < 0.001 compared to *Abcb1a/1b;Abcg2^-/-^* mice. Statistical analysis was applied after log-transformation of linear data.

**Table 3 pharmaceuticals-14-01087-t003:** Plasma and organ pharmacokinetic parameters of selpercatinib in male wild-type and *Abcb1a/1b; Abcg2^−/−^* mice over 2 h after oral administration of 10 mg/kg selpercatinib with or without elacridar.

Parameter	Genotype/Groups			
	Vehicle		Elacridar	
	Wild-Type	*Abcb1a/1b;Abcg2^−/−^*	Wild-Type	*Abcb1a/1b*;*Abcg2^−/−^*
AUC_0–2h_, ng/mL*h	14,092 ± 2816	15,098 ± 1503	12,928 ± 3121	17,294 ± 1513 #
Fold change AUC_0–2h_	1.0	1.1	0.92	1.2
C_max_, ng/mL	8739 ± 1560	8865 ± 792	7466 ± 1848	9617 ± 776 #
T_max_, h	0.75 ± 0.27	1.0 ± 0.0#	0.50 ± 0.27	0.79 ± 0.33
C_brain_, ng/g	210 ± 40	4726 ± 638 ***###	2765 ± 851 ***	4713 ± 484 ***###
Fold increase C_brain_	1.0	22.5	13.2	22.4
Brain-to-plasma ratio	0.041 ± 0.009	0.68 ± 0.18 ***###	0.46 ± 0.03 ***	0.54 ± 0.06 ***
Fold increase ratio	1.0	17	11.5	13.5
C_Liver_, ng/g	18,134 ± 2604	21,445 ± 2330	20,028 ± 4435	23,628 ± 4827
Fold increase C_liver_	1.0	1.2	1.1	1.3
Liver-to-plasma ratio	3.5 ± 0.8	3.1 ± 0.7	3.4 ± 0.3	2.7 ± 0.4
Fold change ratio	1.0	0.89	1.0	0.77
C_SI + SIC_, ng/g	37,604 ± 13,607	18,845 ± 3628 **	26,630 ± 6802	21,226 ± 929 *
Fold increase C_SI + SIC_	1.0	0.50	0.71	0.56
SI + SIC-to-plasma ratio	7.0 ± 1.9	2.7 ± 0.8 ***##	4.6 ± 1.3 *	2.4 ± 0.2 ***##
Fold change ratio	1.0	0.39	0.66	0.34
C_testis_, ng/g	788 ± 159	7020 ± 664 ***	5709 ± 1744 ***	9202 ± 856 ***###
Fold increase C_testis_	1.0	8.9	7.2	11.7
Testis-to-plasma ratio	0.15 ± 0.03	1.0 ± 0.2 ***	0.94 ± 0.08 ***	1.1 ± 0.05 ***
Fold change ratio	1.0	6.7	6.3	7.3

Data are given as mean ± S.D. (n = 6). AUC_0–2h_, area under the plasma concentration–time curve; C_max_, maximum concentration in plasma; T_max_, time point (h) of maximum plasma concentration; C_brain_, brain concentration; C_liver_, liver concentration; SI, small intestine (tissue); SIC, small intestine contents; C_SI+SIC_, small intestine tissue together with small intestine contents concentration; C_testis_, testis concentration; *, *p* < 0.05; **, *p* < 0.01; ***, *p* < 0.001 compared to vehicle-treated wild-type mice; #, *p* < 0.05; ##, *p* < 0.01; ###, *p* < 0.001 compared to elacridar-treated wild-type mice; ^, *p* < 0.05; ^^, *p* < 0.01; ^^^, *p* < 0.001 compared between vehicle-treated *Abcb1a/1b;Abcg2^−/−^* and elacridar-treated *Abcb1a/1b;Abcg2^−/−^* mice. Statistical analysis was applied after log-transformation of linear data.

**Table 4 pharmaceuticals-14-01087-t004:** Plasma and organ pharmacokinetic parameters of selpercatinib in female wild-type, *Cyp3a^-/-^* and Cyp3aXAV mice over 8 h after oral administration of 10 mg/kg selpercatinib.

Parameter	Genotype		
	Wild-Type	*Cyp3a^-/-^*	Cyp3aXAV
AUC_0–8h_, ng/mL*h	52,251 ± 6922	72,243 ± 5642 ***	45,755 ± 3460 ###
Fold change AUC_0–8h_	1.0	1.4	0.88
C_max_, ng/mL	8556 ± 1299	12,295 ± 1311 ***	9477 ± 548 ##
T_max_, h	1.5 ± 1.3	1.4 ± 1.3	0.29 ± 0.10 *#
C_brain_, ng/g	82.4 ± 16.8	86.2 ± 24.2	51.0 ± 15.2 *##
Fold change C_brain_	1.0	1.0	0.62
Brain-to-plasma ratio	0.021 ± 0.002	0.022 ± 0.002	0.022 ± 0.002
Fold change ratio	1.0	1.0	1.0
C_Liver_, ng/g	8255 ± 1615	7896 ± 1274	4386 ± 755 ***###
Fold change C_liver_	1.0	1.0	0.53
Liver-to-plasma ratio	2.1 ± 0.3	2.1 ± 0.2	2.0 ± 0.2
Fold change ratio	1.0	1.0	1.0
C_SI_, ng/g	7302 ± 1116	6384 ± 930	5030 ± 347 ***#
Fold change C_SI_	1.0	0.87	0.69
SI-to-plasma ratio	1.9 ± 0.4	1.7 ± 0.3	2.3 ± 0.6
Fold change ratio	1.0	0.89	1.2
C_SIC_, ng/g	83,866 ± 51,955	44,922 ± 15,127	48,515 ± 13,654
Fold change C_SIC_	1.0	0.54	0.58
SIC-to-plasma ratio	20.0 ± 7.9	11.5 ± 3.3 *	22.3 ± 7.5 ##
Fold change ratio	1.0	0.58	1.1
SIC percentage of dose, %	2.5 ± 0.9	1.4 ± 0.6 *	1.5 ± 0.5
Fold change %	1.0	0.56	0.60

Data are given as mean ± S.D. (n = 6–7). AUC_0–8h_, area under plasma concentration–time curve; C_max_, maximum concentration in plasma; T_max_, time point (h) of maximum plasma concentration; C_brain_, brain concentration. C_liver_, liver concentration; SI, small intestine (tissue); C_SI_, small intestine tissue concentration; SIC, small intestine contents; C_SIC_, small intestine contents concentration; *, *p* < 0.05; **, *p* < 0.01; ***, *p* < 0.001 compared to wild-type mice; #, *p* < 0.05; ##, *p* < 0.01; ###, *p* < 0.001 compared between *Cyp3a^-/-^* and Cyp3aXAV mice. Statistical analysis was applied after log-transformation of linear data.

## Data Availability

Data is contained within the article and Appendix A.

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
