# Peer review of "P-Glycoprotein (ABCB1/MDR1) and BCRP (ABCG2) Limit Brain Accumulation and Cytochrome P450-3A (CYP3A) Restricts Oral Exposure of the RET Inhibitor Selpercatinib (RETEVMO)"

_pharmaceuticals, 2021, doi:10.3390/ph14111087_

Round 1

Reviewer 1 Report

See attachment.

Reviewer 2 Report

The manuscript describes in vitro cell line and in vivo animal model studies investigating the transport and metabolism of the RET inhibitor selpercatinib. The document is well presented, and overall the work is well planned and executed. 

Recommended minor changes

  • As currently drafted, the manuscript does not fully meet ARRIVE recommended guidelines e.g. while the authors state that "All experimental animal protocols were evaluated and approved by the institutional animal care and use committee", they do not name the committee (or which of the authors' institutions it belonged to), nor do they provide an approval reference number. Missing information should be included to fully meet these guidelines prior to publication.
  • The data clearly demonstrates species specificity in selpercatinib transport, however its implications for the applicability of the mouse models to humans are not discussed in section 3 (Discussion).  This should be rectified prior to publication.

Reviewer 3 Report

In my opinion this is a high quality paper, presentation of data is appropriate and the discussion is correct.

I have only a minor comment - You have found that there is no sign of transport of selpercatinib in hABCG2-MDCKII assay contrary to mouse Abcg2-MDCKII assay. This presentation may require a positive control (although the assay is well-known and these are stable cell lines), but it would be more convincing if at least stated in the method, that a known substrate was checked. I think there is a chance that this compound interacts with human ABCG2 too, but it behaves rather as an inhibitor and not a well-transported substrates, other type of assays may help to clarify this point. It would be interesting to know although it does not alter the main message of the paper.

(At row 112 there is a typing error)
